# Surveillance of Drinking Water Quality Worldwide: Scoping Review Protocol

**DOI:** 10.3390/ijerph19158989

**Published:** 2022-07-24

**Authors:** Rayssa Horacio Lopes, Cícera Renata Diniz Vieira Silva, Pétala Tuani Cândido de Oliveira Salvador, Ísis de Siqueira Silva, Léo Heller, Severina Alice da Costa Uchôa

**Affiliations:** 1Postgraduate in Collective Health, Federal University of Rio Grande do Norte, Natal 59064-630, Brazil; isis.siqueira.176@ufrn.edu.br; 2Technical School of Health of Cajazeiras, Federal University of Campina Grande, Cajazeiras 58900-000, Brazil; cicera.renata@professor.ufcg.edu.br; 3School of Health, Federal University of Rio Grande do Norte, Natal 59078-970, Brazil; petala.salvador@ufrn.br; 4René Rachou Institute, Oswaldo Cruz Foundation, Belo Horizonte 30190-009, Brazil; leo.heller@fiocruz.br; 5Public Health Departament, Federal University of Rio Grande do Norte, Natal 59078-900, Brazil; alice.costa@ufrn.br

**Keywords:** drinking water, potable water, public health surveillance, quality control, government

## Abstract

Universal access to clean and safe drinking water is essential for life maintenance since exposure to poor quality water is harmful to health. Drinking water quality is part of public health actions and, together with sanitation, a human right essential for life and a sustainable development goal. Moreover, an independent surveillance system conducted by the Ministry of Health or government agencies is needed for the safety of drinking water quality. We propose a scoping review protocol to identify and map worldwide surveillance actions and initiatives of drinking water quality implemented by government agencies or public health services. This scoping review protocol is based on the Joanna Briggs Institute manual and guided by the PRISMA-ScR. Articles, theses, dissertations, and official documents consulted in the following databases will be included: Medline/PubMed, Scopus, LILACS, Web of Science, Embase, Engineering Village, and gray literature. No date limit or language will be determined. The authors will develop a worksheet for data extraction. Quantitative (simple descriptive statistics) and qualitative data (thematic analysis) will be analyzed. The final scoping review will present the main findings, impacts, challenges, limitations, and possible research gaps related to surveillance of drinking water quality on population health.

## 1. Introduction

Access to clean and safe drinking water (i.e., potable water) is a health, social, and human right issue essential for life maintenance since exposure to poor quality water is harmful to health [1,2,3]. Domestic water use includes ingestion, food preparation, and personal hygiene; therefore, adequate, safe, and accessible water must be available to the entire population [1].

Drinking water quality is part of public health actions and, together with sanitation, a human right essential for life [1,2,3,4,5]. International standards published by the World Health Organization (WHO) since 1958 to assure safe water supply were transformed into guidelines for potable water quality. These guidelines include minimum requirements for water safety and are fundamental for national authorities to elaborate regulations for health protection, considering local environmental, social, economic, and cultural conditions [1].

National and international legislation regarding standards and guidelines for water potability can also derive from national experiences, such as The Safe Drinking Water Act in the United States of America, Ordinance No. 5/2017 in Brazil, European Union Council Directive 98/83/EC, and South African Water Quality Guidelines [3,6].

Access to water services and water quality are different among countries, being less improved in low- or middle-income countries with large rural areas due to the low monitoring capacity [7,8]. Thus, data regarding water quality are scarce or absent in those contexts, increasing the health risks due to unknown water conditions [9].

The WHO estimated that the world population improved the use of safely managed water supply services from 70% to 74% in the last 20 years. However, coverage is still low in rural populations. Therefore, to reach universal access to safe water supply by 2030, as proposed in the Sustainable Development Goals no. 6, the annual progress rate must increase 4-fold for the global population, 10-fold for least developed countries, and 23-fold in fragile contexts [8].

A recent systematic review identified, analyzed, and interpreted more than 11 water quality indexes for human consumption, including pH, nitrate, turbidity, chloride, and sulfate [7]. Microbiological contamination, chemical contaminants, health aspects, and the acceptability of consumers are concerns related to water quality in developing countries [2]. Despite this, part of the population of sub-Saharan Africa does not have access to water quality. For example, only 57% of Ethiopians have access to potable water, increasing diseases transmitted by or linked to water. Ethiopia presents more than 30% of spring water sources contaminated by total or fecal coliforms, requiring monitoring, regular inspection, and appropriate disinfection [4]. China has also faced water-related problems in the last years due to quantitative or qualitative scarcity of potable water. This scarcity is due to the increasing demand for water use, unequal distribution of hydric resources, and water pollution [10].

Three requirements are needed to ensure the safety of potable water: structure of goals, standards, and legislations for water (established by health authorities); adequate and well-managed systems; and independent surveillance, conducted in most countries by the Ministry of Health (or public health) and local and regional structures. Therefore, this surveillance should complement quality control and contribute to public health protection by promoting quality, quantity, accessibility, coverage, purchasing power, and continuity of drinking water supply [1]. Thus, surveillance contributes to achieving the human right to available, accessible, safe, acceptable, and affordable drinking water [5,11].

Health surveillance is essential for public health, policy changes, new programs and interventions, improvement of communication, research assessment, and allocation of investments [12]. Regarding the surveillance of drinking water, well-succeeded programs report data periodically, describe water quality, highlight concerns and priorities in public health, and disseminate results to interested parties [13].

The complex governance in water management and the variety of standards, parameters, and indexes [9] may lead to different surveillance actions of drinking water worldwide. In this sense, the local reality and regional characteristics must guide the adoption of standards.

Furthermore, knowledge of worldwide surveillance actions and initiatives of drinking water quality implemented by government agencies or public health services are needed to systematize evidence, assess surveillance actions, and direct public policies for water quality surveillance. No reviews or research protocols with a similar thematic were found in preliminary research conducted in June 2021 on electronic databases (i.e., Joanna Briggs Institute [JBI] Evidence Synthesis, The Cochrane Library, PROSPERO, and Medline).

In this sense, this study proposes a scoping review protocol to identify and map worldwide surveillance actions and initiatives of drinking water quality implemented by government agencies or public health services.

## 2. Materials and Methods

This is a scoping review protocol based on JBI criteria and the theoretical framework proposed by Arskey and O’malley [14] and updated by Levac, Colquhoun, O’Brien [15], and Peters et al. [16]. This protocol was also guided by the Preferred Reporting Items for Systematic Reviews and Meta-Analyses Extension for Scoping Reviews (PRISMA-ScR) [17]. A protocol was elaborated and registered on the Open Science Framework and can be accessed using the following link (https://doi.org/10.17605/OSF.IO/GBNTP accessed on 22 June 2022).

This review will be conducted in nine stages [16], as shown in Figure 1.

### 2.1. Stage 1. Definition and Alignment of Research Objectives and Questions

The research question was formulated using the PCC mnemonic (Population, Concept, Context), which enables to map a wide range of information to identify possible knowledge gaps, present key concepts, quantify aspects of interest, and expose practices and evidence of a thematic [16]:P-Individuals using potable water via supply servicesC-Surveillance of drinking water qualityC-Government agencies or public health services or both.

Therefore, the research questions are, (1) what are the worldwide surveillance actions and initiatives of drinking water quality implemented by government agencies or public health services? (2) what are the impacts and results of the implemented drinking water quality surveillance actions? and (3) what are the challenges and limitations for developing drinking water quality surveillance actions? The reference concepts of mnemonic elements adopted in this review are shown in Table 1.

### 2.2. Stage 2. Development and Alignment of Inclusion Criteria

Studies regarding surveillance actions on drinking water quality implemented in countries and published in full as research articles, theses, dissertations, or official documents will be included.

We will exclude duplicate publications, literature reviews, theoretical essays, editorials, expert opinions, and publications regarding surveillance or control actions for water not suitable for human consumption.

### 2.3. Stage 3. Description of Evidence Selection

The search strategy will be conducted in three steps to reach the largest number of publications and grey literature.

#### 2.3.1. First Step: Identification of Descriptors and Keywords

The initial search was conducted in PubMed using Medical Subject Headings (MeSH) in English to identify main descriptors, synonyms, and keywords included in titles, abstracts, and indexed terms of publications regarding the thematic. A similar search was conducted in Portuguese using the Virtual Health Library (VHL) and *Descritores em Ciências da Saúde* (DeCS).

Moreover, a librarian improved the search strategy using four controlled vocabularies (three from the health field and one from engineering) (DECS, MESH, EMTREE, and THESAURUS ENGINEERING VILLAGE) to obtain a wide range of multidisciplinary results in different databases. Natural language (not-controlled vocabulary) was also used to increase the sensitivity of the strategy [20,21].

The search strategy was constructed using the Extraction, Conversion, Combination, Construction, and Use model, which enables the development of highly sensitivity search strategies by following a set of complementary steps [20]. Table 2 shows the conversion of mnemonic elements into main keywords.

The complete search strategy constructed for Medline/PubMed is shown in Table A1.

#### 2.3.2. Second Step: Database Definition for Data Collection

After defining a high-sensitive search strategy, data collection will be conducted in Medline/PubMed, Scopus, Web of Science, Embase, LILACS (via VHL), and Engineering Village databases. The latter database from engineering areas and its respective terms were adopted due to the multidisciplinarity of the topic; therefore, studies related to the objectives of this review can be identified. Searches will also be conducted in the following grey literature: Google Scholar, Digital Library of Theses and Dissertations, *Catálogo de Teses & Dissertações*—CAPES, Open Access Theses, and Dissertations, and ProQuest Dissertations & Theses Global.

#### 2.3.3. Third Step: Search for Additional Sources in References of Selected Publications

References of included articles will also be checked to track eligible studies. If needed, corresponding authors will be contacted via e-mail for additional information.

### 2.4. Stage 4. Search for Evidence

Initially, data collection will be conducted using an adequate strategy for each database, and a databank will be created in the Rayyan software free version (Qatar foundation, Doha, Qatar) [22]. Duplicates will be removed, and a pilot test will be conducted with two reviewers (RHL and CRVD). In this pilot test, the titles and abstracts of a random sample comprising 25 studies will be evaluated to verify inclusion criteria and a minimum agreement of 75% [17].

After the pilot test, titles and abstracts of all identified studies will be independently evaluated according to inclusion criteria by two blinded reviewers (RHL and CRVD) using the Rayyan software (Qatar foundation, Doha, Qatar) [22]. Divergences between evaluators will be discussed for consensus. In case of disagreement, a third reviewer (PTCOS) will be consulted.

### 2.5. Stage 5. Selection of Evidence

Publications selected by title and abstract will be fully retrieved and independently extracted by two reviewers after full-text reading. Reasons for exclusion will be highlighted if needed.

Study selection, eligibility criteria, and reasons for inclusion and exclusion in each stage will be reported in a specific flowchart, according to PRISMA-ScR [17]. At this stage, the authors will perform a new search in all databases to verify if new studies can be included.

### 2.6. Stage 6. Extraction of Evidence

Data will be extracted using a worksheet constructed in Microsoft Excel^®^ (Table 3).

Two reviewers trained to extract data will map surveillance actions using geographical identification of where the study was developed. The map will be developed using the GeoDa software, 1.20 version (Center for Spatial Data Science, Chicago, IL, USA).

### 2.7. Stage 7. Analysis of Evidence

This review will generate quantitative and qualitative data that will be analyzed according to proper techniques for each type of data. Quantitative data will be synthesized and analyzed using simple descriptive statistics and absolute and relative frequencies.

In order to analyze the structure and organization of the text, the variables extracted from surveillance actions of drinking water quality and their impacts and results will be analyzed using the Interface de R pour les Analyses Multidimensionnelles de Textes et of Questionnaires (IRaMuTeQ) [23]. Alternatively, qualitative data will be analyzed using thematic analysis as a flexible method to identify meanings and patterns to answer the research question [24,25].

### 2.8. Stage 8. Presentation of Results

The final report guided by the PRISMA-ScR will include the results in flowcharts, charts, or figures [17].

In this stage of the review, summarized results will be presented to five stakeholders (selected according to expertise in surveillance of drinking water quality) and researchers in the field. This stage will be structured to improve the strength of the review and favor socialization and transference of knowledge to individuals interested in the field. This transference will share preliminary results, favor knowledge exchange and consultation of new evidence or research field not found in the review and discuss strategies for disseminating results [15].

Preliminary results and informed consent will be included in an electronic form and sent to stakeholders via e-mail. Stakeholders will not be identified, and authors will request the appreciation of results and possible new fields or evidence. This stage was approved by the research ethics committee of Onofre Lopes University Hospital (no. 4.816.314 and CAAE 47769421.1.0000.5292) on 30 June 2021.

### 2.9. Stage 9. Summary of Evidence, Conclusions, and Implications of Findings

In this stage, a summary of results linked to the aims of the study will be elaborated. Moreover, knowledge gaps will be highlighted for future studies, such as systematic reviews.

## 3. Discussion

This protocol will guide a scoping review to identify and map worldwide surveillance actions and initiatives of drinking water quality implemented by governments or public health authorities. These actions comprise the supervision of regulated and unregulated water services, information to the population served by unregulated supply, consolidation of different information on the general situation of supply, the definition of public health-oriented policies, and participation in the investigation of outbreaks of water-related diseases [13].

The protocol was developed by the research team with knowledge and methodological experience in scoping reviews and surveillance of water quality. A librarian also helped the elaboration of a highly sensitive search strategy based on the combination of four controlled vocabularies to amplify results and enable extensive access to literature, mainly because the search will not be limited by date or language. All information in this protocol (e.g., aim, elements that will formulate the research question, eligibility criteria, methodological stages, and search strategy) favor its transparency; therefore, allowing methodological replicability according to the principles of open science [26] and reducing the risk of bias and unnecessary data duplication.

The results will be reported according to the PRISMA-ScR [17] and summarized to understand the worldwide surveillance actions developed in different contexts, the impacts on population health, challenges, limitations, and possible research gaps that may guide future research. Moreover, the results might be useful for public health professionals, managers, policymakers, and the general population.

As a limitation of the study, we will not search institutional websites of all countries (grey literature) since they are not feasible for the development of this review and because of the extension of the search strategy. In this sense, we will select broad and essential databases for this study, including other sources of grey literature.

## 4. Conclusions

This study protocol presents the characteristics and methodological stages that will guide the scoping review to identify and map worldwide drinking water surveillance implemented by government agencies or public health authorities. The study may also reveal the possible risks in consuming water of poor quality, intervention measures performed by authorities, and efforts and challenges of countries to ensure access to sufficient drinking water of quality.

The results of this review will create socialization with stakeholders and be published in peer-reviewed open-access journals, favoring the dissemination of knowledge within the scientific community. Changes in this protocol will be appropriately reported in the final publication, including dates and justifications.

## Figures and Tables

**Figure 1 ijerph-19-08989-f001:**
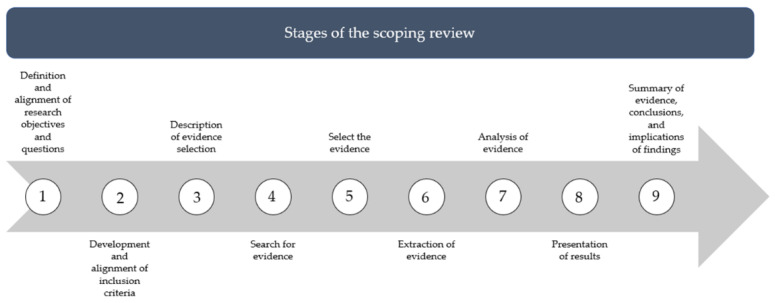
Stages of the scoping review.

**Table 1 ijerph-19-08989-t001:** Definition of concepts to be used in the review.

	Concept	Definition
Individuals using potable water via supply services	Drinking water	Potable water for ingestion, food preparation and production, and personal hygiene, independent of water origin [1,18].
	Potable water	Water for ingestion that attends current potability standards without offering health risks [18].
Surveillance of drinking water quality	Surveillance in public health	Systematic collection, analysis, and interpretation of health-related data to prevent or control diseases or identify important uncommon events in public health, followed by dissemination and use of information for public health actions [19].
	Surveillance of drinking water quality	Investigative activity to identify and evaluate potential health risks related to potable water, contributing to the protection of public health and promotion and improvement of quality, quantity, accessibility, coverage, purchasing power, and continuity of potable water supply. Surveillance authorities determine if a water supplier is fulfilling obligations. In most countries, the Ministry of Health (or public health) and its regional office or departments are responsible for the surveillance of potable water from supply services [1]. Set of actions regularly adopted by the public health authority to verify whether water potability attends to the current legislation (considering socioenvironmental aspects and local reality) and evaluate whether drinking water presents risks for human health [18].
	Quality control	A system to verify and maintain the desired quality level of a product or process via careful planning, adequate equipment use, continuous inspection, and corrective action.
Government agencies or public health services or both	Government	Complex of political institutions, laws, and manners used to govern a specific political unit.

**Table 2 ijerph-19-08989-t002:** Conversion of adopted mnemonic elements.

Mnemonic	Extraction	Conversion
Population	Individuals using drinking water via supply services	Drinking Water Potable Water
Concept	Surveillance of quality	Public Health Surveillance Monitoring Quality Control
Context	Government agencies or public health services	Government Health Care Organization

**Table 3 ijerph-19-08989-t003:** Instruments for data extraction.

Variable	Standardization
Type of material	If article, dissertation, thesis, or official documents
Year of publication	Year of publication
Publication context	Place where the study was conducted (country)
Academic degree of the author	Academic degree of the first author
Aim	Aims of the study
Type of research	Type of research described by authors
Surveillance action	Highlight surveillance action of drinking water quality by publication
Responsible for surveillance action	Highlight who was responsible for the surveillance action (e.g., government agency, sector, company, or official institution)
Impacts or results	Highlight the impacts or results of surveillance actions for collective health (e.g., reduction of diarrheal diseases and improvement in water quality and quantity)
Challenges or limitations	Highlight possible challenges or limitations for surveillance actions

Source: elaborated by authors, 2022.

## Data Availability

Not applicable.

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
