# Peer review of "Surveillance of Drinking Water Quality Worldwide: Scoping Review Protocol"

_ijerph, 2022, doi:10.3390/ijerph19158989_

Round 1
Reviewer 1 Report
The authors have addressed most of the comments given in the previous review. The manuscript reads better now. The paper presents characteristics and methodological stages that will guide the scoping review to identify and map worldwide drinking water surveillance implemented by government agencies or public health authorities. The study may also reveal the possible risks in consuming water of poor quality, intervention measures performed by authorities, and efforts and challenges of countries to ensure access to sufficient drinking water of quality. Specific suggestion:
(1) The author mentioned, the scoping review protocol will be conducted in nine stages. What is the logical relationship or specific process of these nine stages, it is recommended to show it in the form of a diagram, and then introduce the nine stages specifically.
Author Response
Point 1: The author mentioned, the scoping review protocol will be conducted in nine stages. What is the logical relationship or specific process of these nine stages, it is recommended to show it in the form of a diagram, and then introduce the nine stages specifically.
Response: We appreciate the reviewer's suggestion and added a diagram with processes illustrating the nine stages.
An english language review was carried out.

Reviewer 2 Report
This is a study protocol outlining plans for a scoping review of drinking water quality worldwide. The plans are thorough and well thought out. If completed successfully, the scoping review has the potential to be the definitive review article in this field.
Three comments.
The protocol is silent on updates and making the information widely available as circumstances change. An online data base with public access might be worth considering. Excel seems a rather clumsy tool.
Secondly, the protocol is proposing ways to set standards for future measurement and work. I am concerned that much of the information on potable water may exist in the grey literature of consultants and contractors and I wondered how these materials might be included with the more academic articles being trawled? This grey literature is mentioned but the standards used may differ widely.
Thirdly, it would make good sense to geocode the studies so they can be geo-located and perhaps linked to population data bases such as Flowminder and the gridded population figures produced by Andy Tatem and colleagues (https://www.nature.com/articles/s41467-022-29094-x) or by the Columbia remote sensing group.
Author Response
Response to reviewer 2 comments
Point 1: The protocol is silent on updates and making the information widely available as circumstances change. An online data base with public access might be worth considering. Excel seems a rather clumsy tool.
Response: The authors appreciate the suggestion. The scoping review will prioritize the identification and the broad mapping of the evidence regarding the surveillance of drinking water quality worldwide. Thus, data will be updated until stage 5 of the review since there would be no sufficient time to perform new searches and include new data after this phase, according to the JBI manual.
The selected databases (Medline/PubMed, Scopus, LILACS, Web of Science, Embase, and Engineering Village) and gray literature (Google Scholar, Digital Library of Theses and Dissertations, Catálogo de Teses & Dissertações - CAPES, Open Access Theses and Dissertations, and ProQuest Dissertations & Theses Global) can be accessed publicly. Also, the complete database generated in the review will be available for consultation by reviewers or other interested parties in the fighshare repository.
Last, we emphasize that we will use the updated version of the Microsoft Excel spreadsheet to build the database and perform simple descriptive analyses.
Point 2: The protocol is proposing ways to set standards for future measurement and work. I am concerned that much of the information on potable water may exist in the grey literature of consultants and contractors and I wondered how these materials might be included with the more academic articles being trawled? This grey literature is mentioned but the standards used may differ widely.
Response: The authors are grateful for the reviewer's suggestion and concern. However, we aim to identify and map worldwide surveillance actions and initiatives of drinking water quality implemented by government agencies or public health services. Thus, large peer-reviewed and multidisciplinary databases, Google Scholar, and other sources of gray literature are highly recommended for this type of review, according to the methodological framework adopted by the JBI. Moreover, surveillance actions are mainly performed by the public sector; therefore, it is unlikely that there is a gray literature on consultants and contractors.
In this sense, we believe the proposed search strategy will recover the relevant gray literature. Documents regarding the reality of countries and not retrieved by the search strategy will also be selected by manually searching the reference lists of the included studies (i.e., the third step proposed for the scoping review). We inform that this topic was included in the paragraph regarding limitations of the study.
Point 3: It would make good sense to geocode the studies so they can be geo-located and perhaps linked to population data bases such as Flowminder and the gridded population figures produced by Andy Tatem and colleagues (https://www.nature.com/articles/s41467-022-29094-x) or by the Columbia remote sensing group.
Response: The authors are grateful for the reviewer's suggestion and emphasize that they changed the way of preparing the mapping. This will be conducted through geolocation (spatial data and geovisualization) of the publications inserted in the final sample of the review using the GeoDa software. This information was also included in the manuscript.
An english language review was carried out.

This manuscript is a resubmission of an earlier submission. The following is a list of the peer review reports and author responses from that submission.
Round 1
Reviewer 1 Report
The manuscript describes a review protocol to identify and map worldwide surveillance actions and initiatives for drinking water quality. However, it should also demonstrate the applicability of the protocol applying for example to the case of Brazil.
L21: “will be” is repeated
Table 3. Why is the third column empty?
Why is the methodology written in the future tense? The rule is that the methodology is written in the past tense since it was already done.
Table 5 seems not necessary.
The discussion section is too short, which I understand but it is a consequence of the lack of application of the protocol. The manuscript should describe the protocol and discuss its application, giving an example of its application.
Reviewer 2 Report
It would be helpful to provide more information about the geographical scope of the review, beyon "worldwide". Are you going to include absolutely all the countries worldwide? Or are you consider some type of selection criteria? Will country going to be the unit?
Reviewer 3 Report
The manuscript try to propose a scoping review protocol to identify and map worldwide surveillance actions and initiatives of drinking water quality implemented by government agencies or public health services. However, the content and conclusions of the paper are unclear.
Abstract: The abstract should be comprehensive. It should contain brief introduction, objectives, methodology, niche (novelty, research problem) and summary of significant findings. Need to rewrite the abstract. In addition, Research support projects are generally not included in the abstract.
Materials and Methods:The author mentioned, this is a scoping review protocol based on JBI criteria and theoretical framework proposed by Arskey and O’malley. Then, this review will be conducted in nine stages. The author made a general introduction to these nine parts, but did not find the specific content of scientific research, and suggested to explain it with examples, such as what model was selected for data extraction. Methods need to be demonstrated and compared.
Discussion and Conclusion:It is still a general description, and no useful conclusions have been found. What are the characteristics of scoping review protocol? How can it be used to identify and map worldwide surveillance actions and initiatives of drinking water quality implemented by government agencies or public health services?
Overall, I have evaluated this manuscript and do not recommend it to be published in the journal.